

# Land cover and space use influence coyote carnivory: evidence from stable-isotope analysis

Sarah C. Webster[1],*, Joseph W. Hinton[2],*, Michael J. Chamberlain[3], Jazmin J. Murphy[2] and James C. Beasley[1]

[1] Warnell School of Forestry and Natural Resources, University of Georgia, Savannah River Ecology Laboratory, Aiken, South Carolina, United States
[2] Wolf Conservation Center, South Salem, New York, United States
[3] Warnell School of Forestry and Natural Resources, University of Georgia, Athens, Georgia, United States
* These authors contributed equally to this work.

Corresponding author
Joseph W. Hinton, joey@nywolf.org

## ABSTRACT

For many species, the relationship between space use and diet composition is complex, with individuals adopting varying space use strategies such as territoriality to facilitate resource acquisition. Coyotes (*Canis latrans*) exhibit two disparate types of space use; defending mutually exclusive territories (residents) or moving nomadically across landscapes (transients). Resident coyotes have increased access to familiar food resources, thus improved foraging opportunities to compensate for the energetic costs of defending territories. Conversely, transients do not defend territories and are able to redirect energetic costs of territorial defense towards extensive movements in search of mates and breeding opportunities. These differences in space use attributed to different behavioral strategies likely influence foraging and ultimately diet composition, but these relationships have not been well studied. We investigated diet composition of resident and transient coyotes in the southeastern United States by pairing individual space use patterns with analysis of stable carbon ($\delta^{13}$C) and nitrogen ($\delta^{15}$N) isotope values to assess diet. During 2016–2017, we monitored 41 coyotes (26 residents, 15 transients) with GPS radio-collars along the Savannah River area in the southeastern United States. We observed a canopy effect on $\delta^{13}$C values and little anthropogenic food in coyote diets, suggesting $^{13}$C enrichment is likely more influenced by reduced canopy cover than consumption of human foods. We also observed other land cover effects, such as agricultural cover and road density, on $\delta^{15}$N values as well as reduced space used by coyotes, suggesting that cover types and localized, resident-like space use can influence the degree of carnivory in coyotes. Finally, diets and niche space did not differ between resident and transient coyotes despite differences observed in the proportional contribution of potential food sources to their diets. Although our stable isotope mixing models detected differences between the diets of resident and transient coyotes, both relied mostly on mammalian prey (52.8%, SD = 15.9 for residents, 42.0%, SD = 15.6 for transients). Resident coyotes consumed more game birds (21.3%, SD = 11.6 *vs* 13.7%, SD = 8.8) and less fruit (10.5%, SD = 6.9 *vs* 21.3%, SD = 10.7) and insects (7.2%, SD = 4.7 *vs* 14.3%, SD = 8.5) than did transients. Our findings indicate that coyote populations fall on a feeding continuum of

omnivory to carnivory in which variability in feeding strategies is influenced by land cover characteristics and space use behaviors.

# INTRODUCTION

Space use and diet composition are inextricably linked, with animal movement patterns resulting from the diversity and abundance of food resources available for consumption (*Mortelliti & Boitani, 2007*; *Bracis et al., 2015*; *Dupke et al., 2017*). For example, physiological, morphological, and behavioral constraints on mobility can limit an organism's access to food resources, especially in systems where density and distribution of resources vary temporally (*Ydenberg & Krebs, 1987*; *Breuner & Hahn, 2003*; *Höner et al., 2005*; *Shipley, 2007*; *Brandt, Robbins & Bellwood, 2015*). In many species, behavioral adaptations to individual space use such as territoriality have evolved to facilitate resource acquisition, and territoriality can influence foraging success in varying ways (*Kacelink, Houston & Krebs, 1981*; *Waser, 1981*; *Ydenberg & Krebs, 1987*). In some species, energetic trade-offs exist between territorial defense and foraging success, with individuals sacrificing food intake for territorial vigilance (*Kacelink, Houston & Krebs, 1981*; *Ydenberg & Krebs, 1987*). Conversely, territorial defense also can mitigate competition for food resources and lead to increased memory of resource distribution within a territory, thus increasing foraging success within territories (*Bracis et al., 2015*; *Ranc, Cagnacci & Moorcroft, 2022*; *Heathcote et al., 2023*). Such varied outcomes of the same behavioral strategy speak to the complexity of behavioral adaptations to optimize foraging, and show the need to further elucidate the relationship between space use behaviors and diet.

Coyotes (*Canis latrans*) are a generalist carnivore with variable space use and studies commonly report that coyotes exhibit two disparate types of space use, either defending mutually exclusive territories (referred to as residents) or moving nomadically across a landscape as solitary animals (referred to as transients; *Gese, Rongstad & Mytton, 1988*; *Kamler & Gipson, 2000*; *Gehrt, Anchor & White, 2009*; *Hinton, van Manen & Chamberlain, 2015*). However, instead of the traditional binary view of space use by coyotes, recent studies suggest coyotes may exhibit three types of space use (*Morin & Kelly, 2017*; *Sasmal et al., 2019*; *Ellington, Muntz & Gehrt, 2020*). Under a strategy employing three distinct types of space use, transient coyotes can exhibit localized movements while waiting for high quality territories to become available (biding) or move long-distances and rarely reuse sites (transiency). Regardless, resident coyotes typically use less space than transient coyotes and, because of their spatial fidelity, residents have increased access to food resources, thus increasing foraging success to compensate for the energetic costs of defending territories (*Gese, Ruff & Crabtree, 1996a*, *1996b*; *Gese, 2001*; *Hinton et al., 2017*; *Ward et al., 2018*). Conversely, transients do not defend territories, and are thought to redirect energetic costs of territorial defense towards extensive movements in search of

mates and breeding opportunities (*Hinton, van Manen & Chamberlain, 2015*). Differences in space use facilitated by different behavioral strategies (*e.g.*, residents defending mates and resources *vs* transients seeking out mates and resources) likely influence foraging and diet composition, but the relationship between coyote space use and diet remains ambiguous.

Coyotes are typically described as opportunistic omnivores that take advantage of easily accessed food (*Fedriani, Fuller & Sauvajot, 2001*; *Cherry et al., 2016*; *Watine & Giuliano, 2017*; *Colborn et al., 2020*). However, extensive use of fruits and other plant material is not a universal characteristic of coyote diets as some populations exhibit mostly carnivorous diets despite the presence of other food sources (*Prugh, 2005*; *Carrera et al., 2008*; *Dowd & Gese, 2012*; *Hinton et al., 2017*; *Cypher et al., 2018*; *Hinton, Rountree & Chamberlain, 2021*). Indeed, in some landscapes where coyotes coexist with wolves, they exhibit similar carnivorous diets as seen in wolves, albeit with less use of ungulates (*Carrera et al., 2008*; *Hinton et al., 2017*). Given that the coyote's current geographic range encompasses nearly all of North America (*Hinton et al., 2019*), it is not surprising that coyote populations experience a wide range of ecological conditions and fall on a feeding continuum of omnivory to carnivory in which coyotes prioritize eating wild mammals (*Ward et al., 2018*; *Jensen et al., 2022*). By prioritizing nutritionally superior mammalian prey and exhibiting prey-switching behaviors, coyotes can defend finite areas while foraging commensurate with the distribution and availability of prey in their territories (*Ward et al., 2018*).

Traditionally, studies investigated coyote diets by examining scat contents (*Schrecengost et al., 2008*; *Swingen, DePerno & Moorman, 2015*; *Hinton, Rountree & Chamberlain, 2021*) and, to a lesser extent, stomach contents (*Gese, Rongstad & Mytton, 1988*; *Watine & Giuliano, 2017*) and direct observations (*Gese, Ruff & Crabtree, 1996a*). Although these methods can provide short term data of what individuals recently ingested, they may provide biased estimates of diet by only describing non-digestible items rather than describing assimilated diet (*Robbins, Schwartz & Felicetti, 2004*; *Newsome et al., 2010*, *2015*). Additionally, when studies use scats as the unit of replication in their analyses to assess coyote diets, sample sizes can be artificially inflated and lead to pseudo-replication when scats are repeatedly and disproportionately collected from some pack territories (*Hurlbert, 1984*; *Hinton et al., 2017*; *Ward et al., 2018*). To address this, some studies used radio-collared coyotes to identify pack territories and sampled scats from within known home ranges to quantify coyote diets (*Hinton et al., 2017*; *Ward et al., 2018*). However, this approach limits sample collection to only resident animals in which the diet breadth of transient coyotes, and differences between resident and transient diet breadth, are not addressed.

One approach to quantify diet breadth and composition of both residents and transients is to couple space use patterns with analysis of stable isotopes of nitrogen ($\delta^{15}$N) and carbon ($\delta^{13}$C) for individual coyotes (*Newsome et al., 2015*). The $^{15}$N/$^{14}$N ratio is generally interpreted as an indicator of trophic level, whereas the $^{13}$C/$^{12}$C ratio, which differs significantly between plants using the $C_3$ or $C_4$ photosynthetic pathway (*Farquhar, Ehleringer & Hubick, 1989*), can be used to determine the relative importance of $C_4$ resources (either native or anthropogenic) (*DeNiro & Epstein, 1978*;

*DeNiro & Epstein, 1981*) in consumer diets. Comparing isotope values among individual consumers allows researchers to understand diet breadth and, when incorporating isotope data from potential food sources, allows for estimation of diet composition (*Bearhop et al., 2004*; *Caut et al., 2006*). Determining diet breadth for a population of interest can be challenging, especially if diet composition changes seasonally like it does for coyotes (*Schrecengost et al., 2008*; *Swingen, DePerno & Moorman, 2015*; *Ward et al., 2018*). While hair is an effective indicator of stable isotope values for terrestrial mammals, it is important to note that hairs only reflect the isotopic composition of an animal's diet during the time period the hair is metabolically active (*i.e.*, growing; *Newsome et al., 2010*). For canid species, hair provides data on diets of individuals since their last molt period, typically the spring/summer months of April–September in North America (*Castelló, 2018*). Thus, stable isotope values from canid hair samples can be thought to represent summer diet breadth.

Stable isotope analysis can facilitate quantification of coyote diets that are difficult to observe through traditional scat analysis, such as transients. For example, *Newsome et al. (2015)* used radio telemetry and stable isotope analysis to assess diets of resident and transient coyotes in Chicago, IL, USA. They observed significant inter-individual variation in diet among coyotes and prey switching between natural and anthropogenic resources, demonstrating that when paired with individual and landscape metrics, stable isotope analysis allows researchers to effectively quantify diet composition for individuals exhibiting different space use behaviors (*i.e.*, resident *vs* transient coyotes). Similarly, we coupled our previous research involving radio telemetry (*Webster et al., 2022*; *Youngmann et al., 2022*) and scat analyses (*Ward et al., 2018*) with stable isotope analysis of nitrogen and carbon to compare diet composition and overlap of resident and transient coyotes in the southeastern United States. Using stable isotope analysis, our objectives were to (1) assess which individual and landscape metrics influence $\delta^{13}C$ and $\delta^{15}N$ in coyotes; (2) estimate and compare diet breadths of resident and transient coyotes; and (3) estimate the proportional contributions of various food sources to coyote diets.

# MATERIALS AND METHODS

## Study area

Coyotes in our study area inhabited and traversed approximately 4,100 km$^2$ of private and public lands across Georgia (Burke, Columbia, Glascock, Jefferson, Lincoln, McDuffie, Richmond, Warren, and Wilkes Counties), and South Carolina (Abbeville, Aiken, Edgefield, Greenwood, Lexington, McCormick, and Saluda Counties) in the southeastern United States (Fig. 1). Coyotes captured in Georgia and South Carolina commonly moved between our respective study areas that straddled the Savannah River, and likely represented one population, leaving one distinct study area that we referred to as the Savannah River study area (SRA). The SRA had mild sub-tropical climate throughout the year. Summers were generally hot and humid with an average high temperature of 20 °C, while winters were mild with an average low temperature of 1 °C (*Ward et al., 2018*).

The SRA landscape contained a mix of forested, early successional, agricultural, and urban land covers. Forest cover was predominately loblolly (*Pinus taeda*) and shortleaf

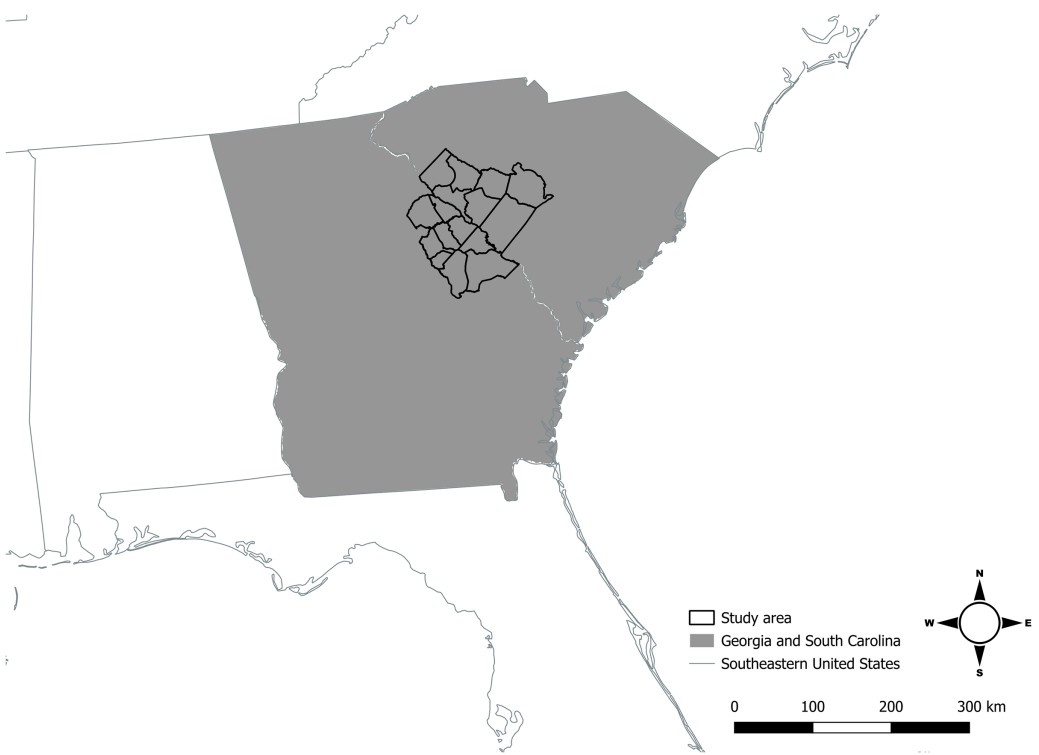

**Figure 1 The 16-county area where Savannah River study area was located in Georgia and South Carolina, United States, in which we sampled coyotes for carbon and nitrogen isotope analyses and estimated coyote space use during 2016–2017.** Source credit: United States Census Bureau. Map created by Joseph W. Hinton using QGIS version 3.32-Lima (*QGIS Development Team, 2023*).

(*Pinus echinata*) plantations, oak (*Quercus* spp.)-hickory (*Carya* spp.)-pine (*Pinus* spp.) woodlands, and successional pine and hardwood forests containing oak, hickory, sweetgum (*Liquidambar styraciflua*), loblolly pine, and shortleaf pine. Agricultural lands were intermittent land cover and consisted mostly of pastures and crops such as cotton (*Gossypium* spp.), corn (*Zea mays*), tobacco (*Nicotiana tabacum*), soybeans (*Glycine max*), peanuts (*Arachis hypogaea*), and peaches (*Prinus persica*).

Using frequency of occurrence of prey in coyote scats, *Ward et al. (2018)* reported that the diets of resident coyotes in SRA consisted predominantly of white-tailed deer (*Odocoileus virginianus*; 42.8%), rabbit (*Sylvilagus* spp.; 28.6%), small mammals (22.3%), fruit (24.1%), and other items (13.3%) such as wild pig (*Sus scrofa*), insects, birds including northern bobwhite (*Colinus virginianus*) and wild turkey (*Meleagris gallopavo*), and anthropogenic food. The small mammal community in SRA included squirrels (*Sciurus* spp.), eastern woodrats (*Neotoma floridana*), hispid cotton rats (*Sigmodon hispidus*), mice (*Peromyscus* spp.), shrews (*Blarina* spp., *Sorex* spp.), voles (*Microtus* spp.), armadillos (*Dasypus novemcinctus*), and Virginia opossum (*Didelphis virginiana*). Fruit common to the SRA and available to coyotes included persimmon (*Diospyros virginiana*), blackberries (*Rubus* spp.), wild plums (*Prunus* spp.), wild grapes (*Vitis* spp.), black cherry (*Prunus serotine*), and peach.

## Animal captures and sampling

To determine space use, we deployed GPS collars on coyotes during fall/winter of 2015–2017. We captured animals with foothold traps (Victor #3 Softcatch, Woodstream Corporation, Lititz, PA, USA; MB 550 or MB 450, Minnesota Trapline Products, Pennock, MN, USA) with offset or padded jaws. Coyotes were typically restrained with a catchpole, muzzle, and hobbles for processing. Although most coyotes were not anesthetized, some were chemically immobilized prior to processing using a Ketamine/Xylazine mixture administered intramuscularly at 0.8 mg/kg for Ketamine and 0.1 mg/kg for Xylazine when it seemed appropriated for the safety of animals and researchers. At the end of procedures, we administered coyotes with Yohimbine at 1.0 mg/kg to intramuscularly as an antagonist to Xylazine. We determined sex and weight of coyotes, and estimated ages of animals by body condition and tooth wear (*Gipson et al., 2000*). To estimate the ages of our study animals, we used April as the birth month for all coyotes because litters were born March–May (*Gier, 1968*; *Knowlton, 1972*). For example, if an animal was captured during January 2016 and presumed to be a pup we estimated the animal's age as 9-months-old assuming it was likely born during April 2015. If the animal was presumed to have been born during April 2014, we estimated its age as 21-months-old. We then categorized coyotes ≥24 months old as adults, 12–24 months old as juveniles, and <12-months-old as pups. We also collected hair samples to determine diet breadth by clipping 10–20 guard hairs from the withers (*i.e.*, area in between the shoulder blades) of coyotes. We stored hair samples in paper envelopes in a climate-controlled facility until the time of analysis.

Prior to release at capture sites, we fitted each coyote with a mortality-sensitive satellite collar (either G2110L Iridium LITE/GPS collar, Advanced Telemetry Systems, Isanti, MN, USA or Litetrack Iridium collar, Lotek Wireless Inc., New Market, Ontario, Canada) set to record locations at 4-h intervals. After release, individuals were monitored for the life of the collar (up to ~18 months) or until mortality. Previously captured individuals were not re-sampled in subsequent years. Our methodology was refined to ensure minimum stress, handling time, and injury to the captured individual, approved by the University of Georgia Institutional Animal Care and Use Committee (protocols A2014 08-025-R2 and A2015 05-004-A5), Georgia Department of Natural Resources (Scientific Collection Permit #29), South Carolina Department of Natural Resources, and Alabama Department of Conservation and Natural Resources, met guidelines recommended by the American Society of Mammalogists (*Sikes & Animal Care and Use Committee of the American Society of Mammalogists, 2016*), and reported in compliance with the ARRIVE guidelines (*Kilkenny et al., 2010*).

## Space use and land cover classification

We used a binary classification of space use (resident or transient) by collared animals instead of a trinary classification (resident, bider, or transient; *Morin & Kelly, 2017*; *Ellington, Muntz & Gehrt, 2020*) to stay consistent with our previous work (*Hinton, van Manen & Chamberlain, 2015*; *Ward et al., 2018*; *Webster et al., 2022*; *Youngmann et al., 2022*). We only classified coyotes with ≥4 months of telemetry data as residents or transients (*Hinton, van Manen & Chamberlain, 2015*; *Ward et al., 2018*). We used fixed

kernel density estimators to calculate coyote space use in R (*R Core Team, 2023*) using the "h-plugin" smoothing parameter in the "ks" package (*Duong, 2007*) to generate 95% isopleths. We classified coyotes as residents when they exhibited stable space use for ≥4 months and had home ranges ≤45 km² (*Hinton, van Manen & Chamberlain, 2015*). For juvenile and adult coyotes, we classified transients as animals with ranging areas >45 km² and unstable, shifting space use (*i.e.*, low site fidelity). However, adult-sized pups (9–11-month-old animals) that were transients during fall/winter captures would have been younger pup-sized animals (0–6 months old) during the previous spring/summer and still dependent on care from breeding residents to have survived spring/summer. Consequently, we classified all pups as residents assuming displacement from natal territories likely occurred during the fall hunting and trapping seasons in which breeders were killed and packs disbanded (*Hinton & Chamberlain, 2022*). This method allowed for unambiguous classification of residents and transients, but also meant that we were unable to determine territorial status of juvenile and adult coyotes with <4 months of movement data due to mortality or collar failure.

Within 95% isopleths, we calculated the percent of agricultural land use, canopy cover, and density of roads to account for the effects of land cover on coyote diets. We determined agricultural land use using the 2016 United States Department of Agriculture (USDA) Forest Service National Land Cover Database (NLCD) by combing cropland (class 81) and pasture (class 82) into one land cover class, whereas mean percent canopy cover was acquired using the 2016 NLCD tree-canopy layer (*Homer et al., 2015*). To calculate percentages of agricultural land use and canopy cover, we used Quantum GIS (QGIS) version 3.32 to overlay resident home ranges and transient ranges against 30-m pixel digital NLCD and NLCD tree-canopy layers. Similarly, we used QGIS to estimate road densities by overlaying home ranges and transient ranges on road layers acquired from the United States Geological Survey (USGS) National Transportation Dataset (NTD) for Georgia (*U.S. Geological Survey, 2024a*) and South Carolina (*U.S. Geological Survey, 2024b*).

## Stable isotope analysis

We prepared hair samples for stable isotope analysis from captured coyotes following the recommendations of *Post et al. (2007)* in which we did not extract lipids from hair samples because a preliminary analysis revealed that C/N values of all samples were <4. This preparation entailed first removing any debris or skin from hair using a sterile surface wipe and then using scissors to macerate hair. We measured 1–2 mg of samples and packed them into pre-combusted 5 × 9 mm tin capsules (Costech Analytical Technologies Inc., Valencia, CA, U.S.A.). Once homogenized, we measured a 1–2 mg sample and packed it into a tin capsule as above. In addition, we prepared duplicate samples for 10% of samples from each species as a quality control measure. We then sent prepared capsules to the University of Georgia Stable Isotope Ecology Laboratory (UGA-SIEL) in Athens, GA, USA for carbon and nitrogen isotope analysis by continuous flow isotope ratio mass spectrometry. Isotope ratio analysis involved transforming solid-phase samples to gas by rapid and complete flash combustion of sample material. The ionized combustion

products were then mass analyzed by means of differing mass/charge ratios among the various isotopic species of $CO_2$ and $N_2$. The UGA-SIEL used standards in each analysis for QA/QC as well as calculating $\delta^{13}C$ and $\delta^{15}N$ (the difference between a sample and natural abundance standard for which $\Delta$ = zero; reported in parts per thousand) with air being the standard for nitrogen analyses and the Chicago PDB Marine Carbonate Standard (U.S. National Institute of Standards and Technology (NIST)), Gaithersburg, MD, U.S.A.) for carbon. The UGA-SIEL used NIST reference materials 8549, 8558, 8568, and 8569 to calibrate working standards in the laboratory. Measurement accuracy was within 0.2‰ for carbon and 0.3‰ for nitrogen.

We used a multi-faceted approach to determine differences in diet breadth between resident and transient coyotes and estimate diet composition based on sampled food resources. First, we used analysis of variance (ANOVA) and *t*-tests to explore differences in $\delta^{13}C$ and $\delta^{15}N$ values among coyote age, sex, and space use status. We also used ANOVAs to explore differences in $\delta^{13}C$ and $\delta^{15}N$ values among our landscape metrics. We then used generalized additive models (GAMs) to assess the effect of residency, age, and body mass of coyotes and the size and land cover composition of space used by coyotes on $\delta^{13}C$ and $\delta^{15}N$ values. We used GAMs rather than standard linear models because GAMs can more efficiently capture nonlinear variation. For our GAMs, we used a thin-plate spline shrinkage approach to modify the smoothing penalty for model selection purposes and restricted *k* to avoid overly complicated smooths. We included a binary variable in our GAMs to represent residency (resident = 1, transient = 0) whereas age (months), space use ($km^2$), body mass (kg), road density ($km/km^2$), canopy cover ($\bar{x}$% cover of home and transient ranges), and agricultural cover ($\bar{x}$% of home and transient ranges) were included as continuous variables. Because we classified all pups as residents, size of space used by coyotes was not correlated with residency ($t_{39}$ = 0.908, $P$ = 0.369) and was treated as a continuum of localized-to-long distance ranging behaviors. For each isotope value ($\delta^{13}C$ and $\delta^{15}N$), we fit the full model with shrinkage and retained all terms for inference (*Marra & Wood, 2011*). The effective degrees of freedom (edf) estimated from GAMs are a proxy for the degree of non-linearity relationships between predictor and response variables (*Marra & Wood, 2011*). Terms with 0 edfs have no effect on model fit, whereas non-significant terms that have positive edfs have small but non-zero effects.

To compare niche structure and overlap between resident and transient coyotes relative to potential food sources, we estimated isotopic niche space for both groups by plotting the isotope values for all coyote hair samples, calculating convex hulls around each group, and then calculating size-corrected standard Bayesian ellipses using package "SIBER" (*Jackson et al., 2011*). We then computed the total area of each convex hull (TA), sample size-corrected standard ellipse area (SEAc), and the proportion of overlap of standard ellipses between resident and transient coyotes. Because we detected an effect of land cover on stable isotope values (see Results), we also compared niche structure and overlap between coyotes on SRA and the Albemarle Peninsula of North Carolina (*Arbogast, Hodge & Brenner-Coltrain, 2017*). Diets, space use, and habitat selection by coyotes on the Albemarle Peninsula are well known (*McVey et al., 2013*; *Hinton, van Manen & Chamberlain, 2015*; *Hinton et al., 2017*) and our familiarity with both populations allows

for an appropriate comparison to draw inferences about coyote ecology in the southeastern United States. When comparing SRA and Albemarle Peninsula coyotes, we included all SRA coyotes with stable isotopic values sampled during our study because meta-data (*e.g.*, age, weight, space used, and months monitored) were not used in our SIBER analysis (Table S1).

Finally, we used stable isotope mixing models to estimate proportional contribution of food items to coyote diets. We targeted food items of coyotes in our study area consumed during spring/summer (*Schrecengost et al., 2008*; *Ward et al., 2018*) to correspond to the time period relevant to our hair samples (*i.e.*, when hair would be metabolically active). These food items included fruit, insects, small mammals including Leporidae, game birds (northern bobwhite and wild turkey), white-tailed deer, and anthropogenic food. We collected potential prey samples opportunistically throughout spring and winter of 2019. We collected white-tailed deer hair samples from legally harvested or road killed individuals in Aiken County, South Carolina, and Oglethorpe and Athens-Clarke Counties, Georgia. Hispid cotton rats (*Sigmodon hispidus*) were opportunistically live trapped using Sherman traps (H.B. Sherman Traps, Inc. Tallahassee, FL, USA) in Athens-Clarke County, Georgia. Hair samples from all prey species were collected by clipping hairs from the withers (*i.e.*, area between the shoulder blades) of each animal. All animal handling procedures for rodents were approved by the University of Georgia Institutional Animal Care and Use Committee (protocol A2018 06-024). Blackberry and persimmon samples were collected on private farms in Oglethorpe County, GA. We stored samples in plastic storage bags, with hair samples stored at room temperature and fruit samples in a −20 °C freezer until analysis. We used the same methodology to homogenize, sample, and pack prey hair into a tin capsule as we did for coyote hair and then shipped capsules to UGA-SIEL for carbon and nitrogen isotope analysis. We homogenized fruit samples using an electric blade grinder. Because our collected prey samples did not encompass all potential prey types consumed by coyotes, we acquired additional isotope values for other potential prey and anthropogenic foods from the literature (Table S2). We relied on published isotopic data for food sources geographically proximate to our study area. However, when isotopic data were not available for food sources within SRA and surrounding areas, we used data from adjacent regions. For example, we used stable isotopic data for eastern cottontails (*Sylvilagus floridanus*) from the Edwards Plateau region of south-central Texas (*Smith et al., 2014*) because such data was not available for the southeastern United States. We then used the average carbon ($\delta^{13}$C) and nitrogen ($\delta^{15}$N) values for each group (*i.e.*, fruit, insects, small mammals, game birds, white-tailed deer, and anthropogenic) to identify potential dietary endpoints (Table 1).

We estimated the proportional contributions of each food source to the composition of coyote diets using the R package "simmr" (*Parnell et al., 2010*) in which we used a Bayesian stable isotope mixing model. The mixing model used Markov chain Monte Carlo (MCMC) methods and accounted for error in estimates of fractionation factors and variation in elemental concentrations of carbon and nitrogen in food sources that could bias model output (*Phillips & Koch, 2002*). We assessed model convergence using Gelman diagnostics within package 'simmr' in which convergence diagnostic values fell between 0.01 of 1.00,

**Table 1 Mean and standard deviation (SD) of $\delta^{13}C$ and $\delta^{15}N$ isotopes for common coyote food sources (Table S2).**

| Food source | $\delta^{13}C$ | | $\delta^{15}N$ | |
|---|---|---|---|---|
| | Mean | SD | Mean | SD |
| Fruit | −27.5 | 1.6 | −1.4 | 3.4 |
| Insects | −23.1 | 5.0 | 0.9 | 0.9 |
| Small mammals | −21.8 | 2.7 | 3.8 | 1.4 |
| Game birds | −21.4 | 3.7 | 5.0 | 0.7 |
| White-tailed deer | −23.6 | 1.2 | 3.7 | 1.1 |
| Anthropogenic sources | −16.7 | 4.3 | 4.3 | 1.7 |

indicating our models converged. A primary challenge in using stable isotope mixing models is that food sources in many study systems may have isotopic signatures that overlap in isotope space, preventing mixing models from discriminating among food sources (*Phillips et al., 2014*). We used correlations from matrix plots of food source proportions to evaluate if mixing models had difficulty separating food sources. Low correlations (0.00–0.39) between sources indicated that the stable isotope mixing models performed well, whereas high correlations (0.60–1.00) indicate when models struggling to differentiate between food sources (*Parnell et al., 2010*).

Prior to analysis, we corrected for changes in atmospheric $CO_2$ over the last ~150 years (*i.e.*, the Suess effect; *Keeling, 1979*). Based on core records, we applied a time dependent correction of −0.022‰ per year (*Francey et al., 1999*). All samples were corrected to reflect 2019 values. Additionally, for fruit, insects, game birds, and anthropogenic food, we applied diet-to-hair trophic discrimination factors (TDFs) of 2.6‰ for $\delta^{13}C$ and 3.4‰ for $\delta^{15}N$ (*Roth & Hobson, 2000*; *Derbridge, Krausman & Darimont, 2012*). For small mammals and white-tailed deer, we used a TDF of 1.6‰ for $\delta^{13}C$ assuming that the hair of prey would be 1‰ lower than the TDF for prey tissues (*e.g.*, muscle and other organs) that would have been assimilated by coyotes (*Newsome et al., 2015*). We did not adjust TDFs for $\delta^{15}N$, because tissue-specific TDFs were not reported for $\delta^{15}N$ (*Caut, Angulo & Courchamp, 2009*; *Newsome et al., 2015*). Finally, we used an error estimate (SD) of 0.5‰ for $\delta^{13}C$ and $\delta^{15}N$ in our mixing models (*Newsome et al., 2015*). We conducted all statistical analyses in Program R (*R Core Team, 2023*).

## RESULTS

We collected guard hair samples from 58 coyotes captured across SRA. Of those, we excluded 16 individuals because we could not adequately determine their space use behaviors (*N* = 12), age (*N* = 2), or both (*N* = 3), leaving 41 individuals included in our analyses. Adults ($\bar{x}_{age}$ = 42.4 months, SD = 12.2) and juveniles ($\bar{x}_{age}$ = 20.9 months, SD = 1.5) accounted for 34.1% and 29.3% of our captured coyotes, respectively, whereas pups ($\bar{x}_{age}$ = 9.3 months, SD = 0.7) accounted for 36.6% of captures (Table S1). The 15 pups (four females, 11 males) monitored with telemetry were between 9 months and 11 months old at the time of capture and, given that they would have been dependent on

breeders for food (*e.g.*, nursing and regurgitation) for some of the previous spring/summer months, we considered all pups as residents despite telemetry data indicating transiency for some ($N = 10$) during their monitoring period. Based on these criteria we classified 26 coyotes as residents (63.3%; 10 females, 16 males) and 15 as transients (35.7%; seven females, eight males; Table S1).

We observed no correlation between $\delta^{13}C$ and age of coyotes ($F_2 = 1.583$, $P = 0.219$), but we detected a negative correlation between $\delta^{15}N$ values and age ($F_2 = 4.224$, $P = 0.022$) in which pups had greater $\delta^{15}N$ values than did juveniles and adults. We observed no correlation between $\delta^{13}C$ ($t_{39} = 0.186$, $P = 0.853$) and $\delta^{15}N$ ($t_{39} = 0.541$, $P = 0.591$) values and sex of coyotes. We observed no correlation between $\delta^{13}C$ values and space use by coyotes ($t_{39} = -1.325$, $P = 0.193$), but we detected a negative correlation between $\delta^{15}N$ values and transiency ($t_{39} = -2.204$, $P = 0.034$).

Most SRA coyotes inhabited three local areas: Thomson-Augusta area in Georgia, Savannah River Site in South Carolina, and Edgefield-Aiken area in South Carolina. These areas differed in percent canopy cover in space used by coyotes ($F_2 = 26.94$, $P \leq 0.001$) in which we detected no difference in canopy cover between the Thomson-Augusta area (68.1%, SD = 8.7) and Savannah River Site (72.7%, SD = 3.2) but observed less canopy cover in the Edgefield-Aiken area (54.7%, SD = 6.8) when compared to the other two areas. We also detected differences in agricultural cover in space used by coyotes ($F_2 = 27.84$, $P \leq 0.001$) in which we detected no difference in percent agricultural cover between the Thomson-Augusta area (13.8%, SD = 7.2) and Savannah River Site (3.5%, SD = 5.3) but observed more agricultural cover in the Edgefield-Aiken area (32.4%, SD = 12.3) when compared to the other two areas. Road density differed in space used by coyotes ($F_2 = 6.27$, $P = 0.004$) in which we detected no difference between the Savannah River Site (0.62, SD = 0.17) and Thomson-Augusta (0.75, SD = 0.14) and Edgefield-Aiken (0.56, SD = 0.13) areas but detected greater road density in the Thomson-Augusta area when compared to the Edgefield-Aiken area. We detected a difference in $\delta^{13}C$ values in coyotes inhabiting the three areas ($F_2 = 6.169$, $P = 0.004$) in which the Savannah River Site (−21.5‰, SD = 1.3) and Thomson-Augusta area (−21.5‰, SD = 1.6) had similar $\delta^{13}C$ values but were more $^{13}C$ depleted when compared to the Edgefield-Aiken area (−19.9‰, SD = 1.4). We also detected a difference in $^{15}N$ values in coyotes inhabiting the three areas ($F_2 = 5.718$, $P = 0.007$) in which the Edgefield-Aiken (7.4‰, SD = 1.4) and Thomson-Augusta (7.0‰, SD = 0.7) areas had similar $\delta^{15}N$ values but were more $^{15}N$ enriched when compared to the Savannah River Site (5.8‰, SD = 1.0).

Our GAM indicated $\delta^{13}C$ values had a negative, non-linear correlation with percent canopy cover in space used by coyotes (Table 2, Fig. 2), whereas the remaining five predictors had non-significant influences on $\delta^{13}C$ values. Additionally, a strong non-linear correlation between $\delta^{15}N$ and percent agricultural cover in which $\delta^{15}N$ values were greatest in space used by coyotes with intermediate agricultural cover (Table 2, Fig. 3).

Adults, juveniles, and pups occupied similar isotopic niche space (Fig. 4A). Pups exhibited 52.4% overlap with adults and 44.1% overlap with juveniles, whereas adults and juveniles exhibited 66.8% overlap. When we estimated Bayesian standard ellipses for age classes, pups had greater estimated SEAc than did adults and juveniles (SEAc = 6.78 for

**Table 2 Detailed summary table of generalized additive model (GAM) results for modeling the effects of individual and landscape metrics on $\delta^{13}C$ and $\delta^{15}N$, and C/N values in coyotes along the Savannah River area of Georgia and South Carolina, USA (2015–2017).**

| Model | Family | Link function | $\bar{R}^2$ | Deviance (%) |
|---|---|---|---|---|
| $\delta^{13}C$[a] | Gaussian | Identity | 0.243 | 39.3 |
| $\delta^{15}N$[b] | Gaussian | Identity | 0.427 | 58.8 |

| | $\delta^{13}C$ | | | | $\delta^{15}N$ | | | |
|---|---|---|---|---|---|---|---|---|
| | β | SE | t value | P | β | SE | t value | P |
| Intercept | −20.901 | 0.469 | −44.550 | <0.001 | 6.511 | 0.351 | 18.540 | <0.001 |
| Resident | 0.504 | 0.653 | 0.446 | 0.446 | 0.773 | 0.496 | 1.560 | 0.130 |
| | Edf | Red.df | F | p-value | edf | Red.df | F | p-value |
| s(Age)[c] | 1.000 | 1.000 | 0.576 | 0.454 | 1.000 | 1.000 | 0.584 | 0.451 |
| s(Body mass)[d] | 1.000 | 2.402 | 0.025 | 0.874 | 2.232 | 2.749 | 1.412 | 0.226 |
| s(Space use)[e] | 1.000 | 1.000 | 2.114 | 0.156 | **1.317** | **1.517** | **2.676** | **0.076** |
| s(% canopy)[f] | **1.912** | **1.000** | **3.898** | **0.026** | 1.146 | 1.248 | 0.191 | 0.827 |
| S(% agriculture)[g] | 1.000 | 1.000 | 2.335 | 0.136 | **3.508** | **4.238** | **2.600** | **0.057** |
| S(Road density)[h] | 1.000 | 1.000 | 0.867 | 0.4674 | **1.000** | **1.000** | **6.939** | **0.013** |

Notes:
Significant effects are shown in bold.
[a] $\delta^{13}C \sim s(Age, k = 7) + s(Body\ mass, k = 7) + s(Space\ use, k = 7) + s(\%\ canopy, k = 7) + s(\%\ agriculture, k = 7) + s(road\ density, k = 7) + resident.$
[b] $\delta^{15}N \sim s(Age, k = 7) + s(Body\ mass, k = 7) + s(Space\ use, k = 7) + s(\%\ canopy, k = 7) + s(\%\ agriculture, k = 7) + s(road\ density, k = 7) + resident.$
[c] Age of coyotes (months).
[d] Body mass of coyotes (kg).
[e] Space use by coyotes ($km^2$).
[f] $\bar{x}$% canopy cover in coyote home and transient ranges.
[g] $\bar{x}$% agricultural cover in coyote home and transient ranges.
[h] Road density ($km/km^2$) in coyote home and transient ranges.

pups, SEAc = 4.55 for adults, and SEAc = 3.49 for juveniles), indicating pups had the broadest isotopic niche space of the group. When comparing resident and transient coyotes, we observed them occupying similar isotopic niche space (Fig. 4B). The overlap between standard ellipses of resident and transient coyotes was 67.1%. Our Bayesian standard ellipses for both groups showed that residents had greater estimated SEAc than did transients (SEAc = 5.91 for residents and SEAc = 4.74 for transients), indicating residents had a slightly broader isotopic niche space. Standard ellipse estimates indicated that coyotes in SRA and on the Albemarle Peninsula of North Carolina occupied different isotopic niche space in which the populations exhibited 14.1% overlap (Fig. 4C). Coyotes in SRA had smaller SEAc (5.16 *vs* 8.45) than did coyotes in northeastern North Carolina indicating that populations on the Albemarle Peninsula had greater niche breadth than those in SRA.

The dietary proportions estimated by our stable isotope mixing models included uncertainty resulting from food sources overlapping in the iso-space plot (Fig. 5). In particular, small mammals and white-tailed deer overlapped significantly in isotope space in comparison to other food sources. Our matrix plots showed moderate negative correlations between small mammals and white-tailed deer across age classes and space use status (Figs. S1–S5). Additionally, we detected strong negative correlations between fruit

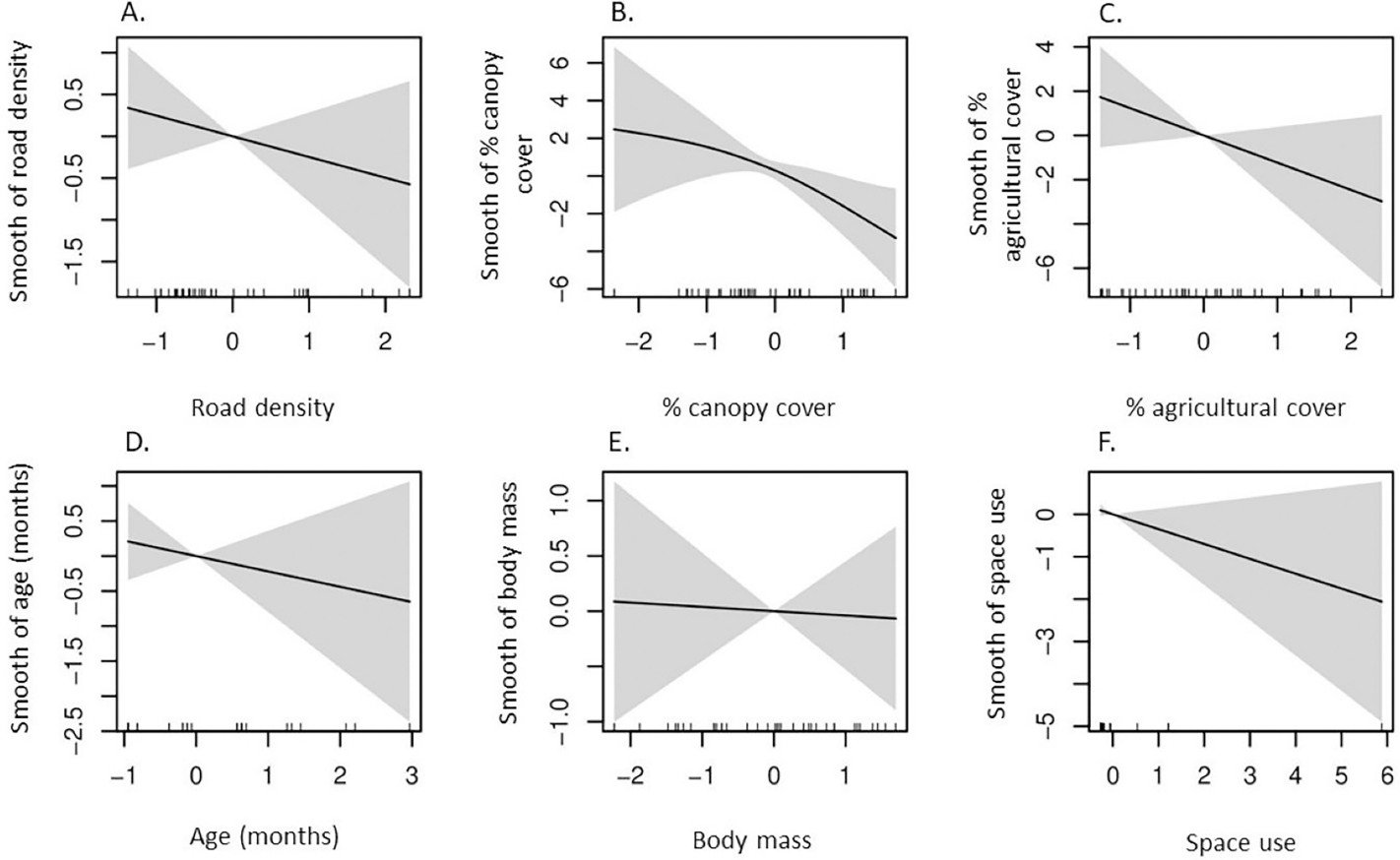

**Figure 2 According to our generalized additive model, partial effects of predictors on δ¹³C values in coyotes of the Savannah River area Georgia and South Carolina, USA, 2016–2017.** Plots represent smooth functions for the mean effect of (A) road density (km/km²), (B) percent canopy cover, (C) percent agricultural cover, (D) age (months) of coyotes, (E) body mass (kg) of coyotes, and (F) space use by coyotes (km²) on δ¹³C values. All predictors were scaled. The gray shaded area represents the 95% confidence intervals for predictors and the rug ticks at the bottom of each plot indicate the coverage of the range of values for each predictor.

and white-tailed deer, two food sources that did not visually appear to overlap in isotope space (Fig. 5, Figs. S1–S5). Negative correlations between food sources implies that when one food source (*e.g.*, white-tailed deer) was consumed at the top of its probability range, it is likely that the other source (*e.g.*, small mammals) was not consumed and the stable isotope mixing model was unable to isolate the contribution of either in isolation (*Parnell et al., 2013*).

The broad 95% credible intervals of food sources also revealed uncertainty in our stable isotope mixing models for age class (Table 3) and space use status (Table 4, Fig. 5). To account for some of the uncertainty, we grouped small mammals and white-tailed deer into a "mammalian prey" category. Most food items consumed by coyotes regardless of age were mammalian prey: adult (44.3%, SD = 15.5), juvenile (44.5%, SD = 15.6), pup (47.1%, SD = 16.7). Fruit, insects, game birds, and anthropogenic food comprised the remainder of coyote diets with pups likely to consume more game birds and less fruit than did adults (Table 3). When mammalian prey was separated into small mammals and white-tailed deer, coyotes exhibited similar use of both prey types regardless of age (Table 3).

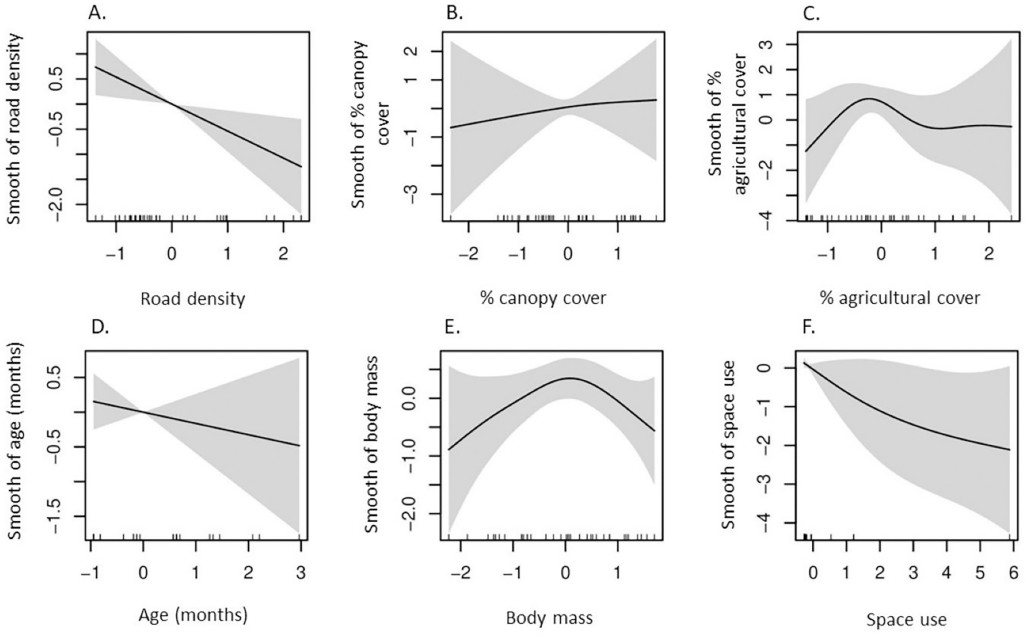

**Figure 3 According to our generalized additive model, partial effects of predictors on δ¹⁵N values in coyotes of the Savannah River area Georgia and South Carolina, USA, 2016–2017.** Plots represent smooth functions for the mean effect of (A) road density (km/km²), (B) percent canopy cover, (C) percent agricultural cover, (D) age (months) of coyotes, (E) body mass (kg) of coyotes, and (F) space use by coyotes (km²) on δ¹⁵N values. All predictors were scaled. The gray shaded area represents the 95% confidence intervals for predictors and the rug ticks at the bottom of each plot indicate the coverage of the range of values for each predictor.            

Most food items consumed by resident and transient coyotes were mammalian prey (Table 4), but residents consumed more mammalian prey (52.8%, SD = 15.9 *vs* 42.0%, SD = 15.6) and game birds (21.3%, SD = 11.6 *vs* 13.7%, SD = 8.8) than did transients. Transients consumed more fruit (21.3%, SD = 10.7 *vs* 10.5%, SD = 6.9) and insects (14.3%, SD = 8.5% *vs* 7.2%, SD = 4.7) than did residents. The contribution of anthropogenic foods to resident and transient diets were similar and approximately 8% (Table 4). When mammalian prey was separated into small mammals and white-tailed deer, residents consumed more white-tailed deer (32.4%, SD = 17.0 *vs* 24.3%, SD = 15.8) than did transients but both consumed similar amounts of small mammals (Table 4).

## DISCUSSION

Despite the substantial overlap in spring/summer diets and niche breadth of resident and transient coyotes, we observed individual and landscape effects on variation of δ¹³C and δ¹⁵N values. Most variability in δ¹³C and δ¹⁵N of SRA coyotes reflected variation in canopy cover (δ¹³C), agriculture (δ¹⁵N), road density (δ¹⁵N), and coyote space use (δ¹⁵N). Despite some uncertainty in our stable isotope mixing models, diets of resident and transient coyotes appeared to be similar and consisted mostly of mammalian prey (small mammals and white-tailed deer) and game birds followed by fruit and insects. Anthropogenic food comprised a small proportion of coyote diets (<11%). Most of the coyotes we sampled hair from were captured and monitored by *Ward et al. (2018)* and our stable isotope analysis
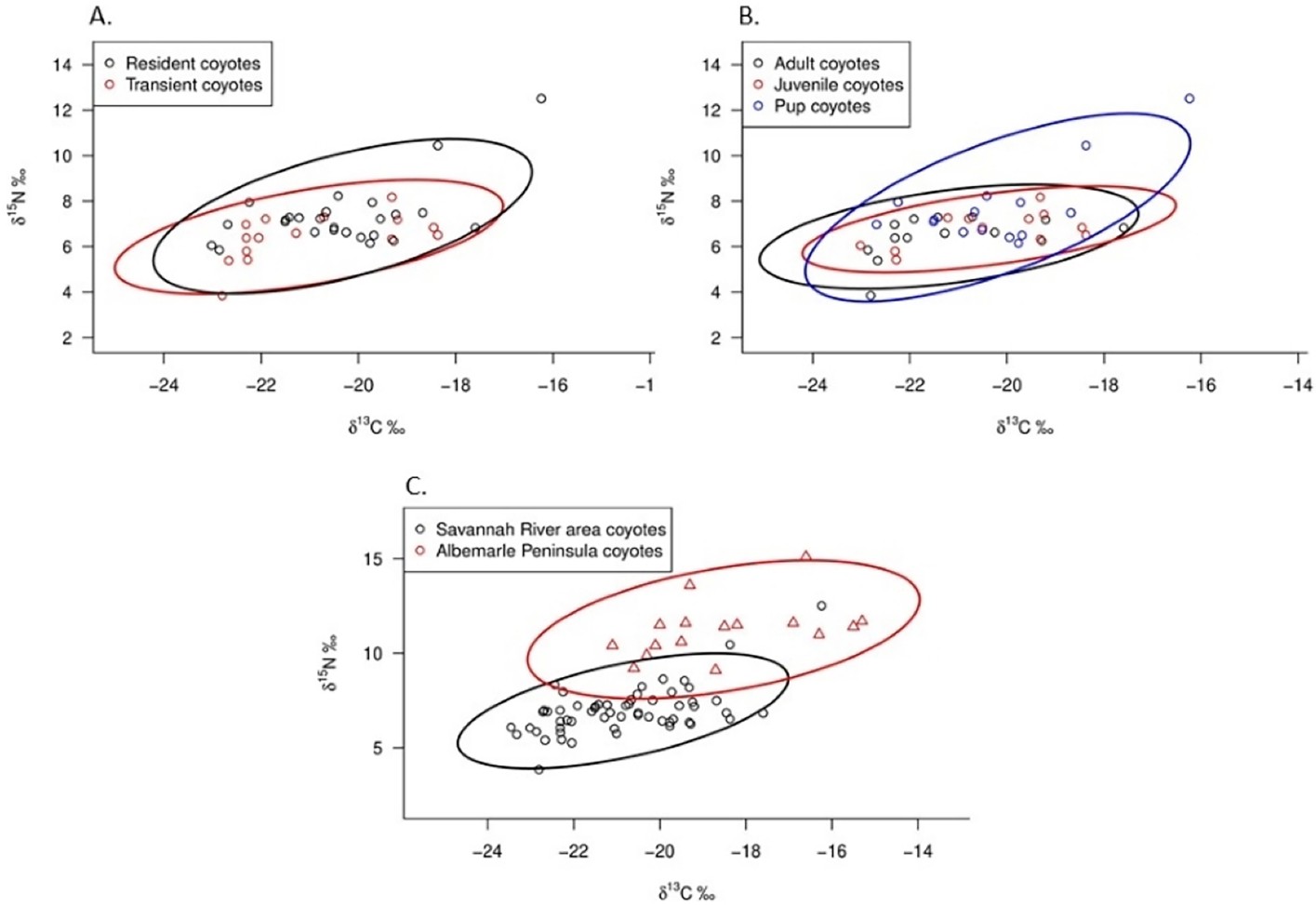

**Figure 4** $\delta^{13}C$ and $\delta^{15}N$ values and estimated Bayesian standard area ellipses (SEAc) of isotopic niche space for (A) coyote age classes and (B) resident and transient coyotes in the Savannah River area of Georgia and South Carolina, U.S during 2016–2017. (C) Comparison.

reflects diets of SRA coyotes during spring/summer prior to *Ward et al.*'s *(2018)* sampling of scats from pack territories. Notably, we observed similar contributions of insects and small mammals to resident diets as did their study but observed smaller contributions of fruit (Table 4). However, we observed significantly less contribution of white-tailed deer and greater contributions of game birds and anthropogenic foods to the diets of resident coyotes than did *Ward et al. (2018)* (Table 4). Although this may indicate potential flaws in using traditional scat analysis to produce accurate estimates of food items, stable isotope mixing models are known to be sensitive to variability in discrimination factors and source materials, and can produce unreliable estimates of diet (*Phillips & Gregg, 2003*; *Parnell et al., 2010*; *Bond & Diamond, 2011*; *Phillips et al., 2014*; *Robinson, Franke & Derocher, 2018*). Therefore, results from our stable isotope analysis and mixing models should be considered in the larger context of existing research on southeastern coyotes to draw reliable conclusions.

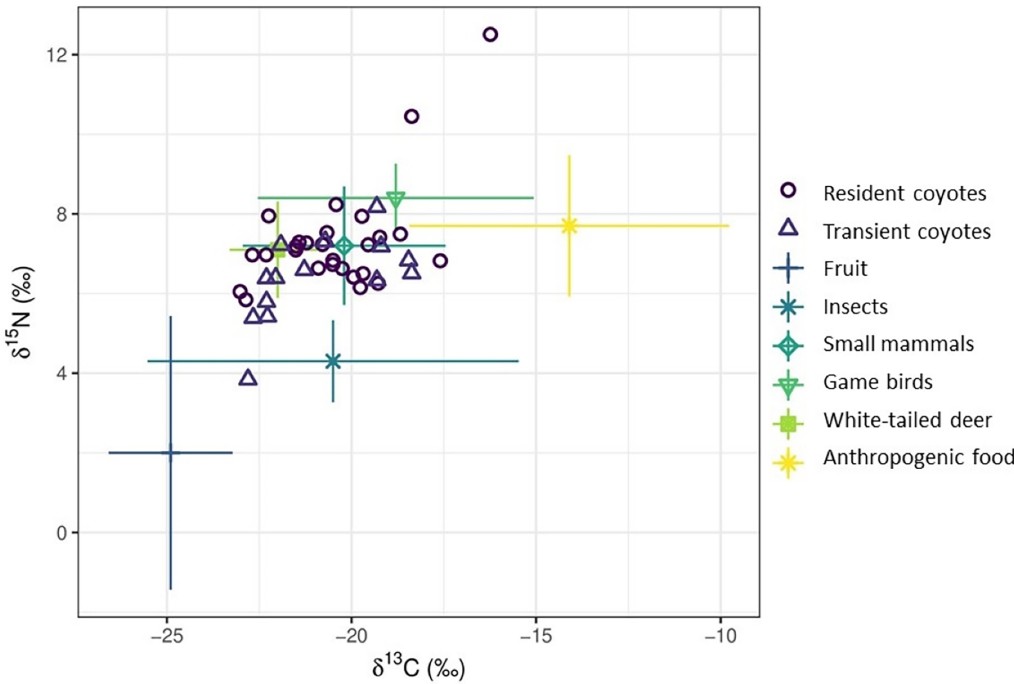

**Figure 5 Comparisons of estimated proportional contributions of fruit, insects, small mammals, game birds, white-tailed deer, and anthropogenic foods to the diet of resident and transient coyotes in the Savannah River Area of Georgia and South Carolina, U.S during.**

**Table 3 Estimates of mean, standard deviation (SD), and 95% credible intervals (CrI) of proportional contributions of potential food sources to diets of adult, juvenile, and pup coyotes from the Savannah River area (SRA) of Georgia and South Carolina, U.S. during 2016–2017.**

| Food source | Adults | | | Juveniles | | | Pups | | |
|---|---|---|---|---|---|---|---|---|---|
| | Mean | SD | 95% CrI | Mean | SD | 95% CrI | Mean | SD | 95% CrI |
| Anthropogenic | 7.6% | 4.5 | 1.4–18.3% | 10.8% | 5.7 | 2.1–23.4% | 9.9% | 5.5 | 1.6–22.0% |
| Fruit | 21.6% | 0.5 | 3.3–41.8% | 14.6% | 8.4 | 2.1–32.8% | 10.8% | 7.4 | 1.3–28.9% |
| Insects | 13.2% | 8.1 | 1.8–32.1% | 12.8% | 7.5 | 2.1–29.8% | 8.6% | 5.9 | 1.2–22.5% |
| Game birds | 13.2% | 8.6 | 1.8–33.3% | 17.3% | 10.3 | 2.4–40.1% | 23.5% | 13.9 | 2.6–52.9% |
| Mammals | 44.4% | 15.5 | 14.7–74.4% | 44.5% | 15.6 | 14.6–73.5% | 47.2% | 16.7 | 14.0–78.6% |
| [a]Small mammals | 17.1% | 12.0 | 1.9–45.2% | 18.8% | 12.7 | 2.4–49.8% | 21.2% | 15.3 | 2.0–58.1% |
| [a]White-tailed deer | 27.3% | 16.5 | 3.2–63.3% | 25.7% | 15.3 | 3.1–58.3% | 26.0% | 16.4 | 2.8–62.2% |

**Note:**
[a] Food sources grouped together based on taxonomy (Class level) as "mammalian prey" because they exhibited considerable overlap in isotope space (Fig. 3).

It is important to note that the movement data used to categorize our coyotes as residents or transients and to estimate space use were collected after animal captures and hair sampling. Consequently, our results reflect the previous spring/summer diets of coyotes prior to their fall/winter captures and it is possible that some sampled individuals transitioned from residency to transiency or *vice versa* between the time period represented by hair samples and the time period in which we monitored movements. Additionally, for transient coyotes, space use may not reflect use of areas from the previous

**Table 4 Estimates of mean, standard deviation (SD), 95% credible intervals (CrI), and 95% confidence intervals (CI) of proportional contributions of potential food sources to diets of resident and transient coyotes from the Savannah River area (SRA) of Georgia and South Carolina, U.S. during 2016–2017.**

| Food source | Residents | | | Transients | | | *Ward et al. (2018)*[a] | | |
|---|---|---|---|---|---|---|---|---|---|
| | Mean | SD | 95% CrI | Mean | SD | 95% CrI | Mean | SD | 95% CI |
| Anthropogenic | 8.2% | 4.2 | 1.7–17.7% | 8.7% | 4.9 | 1.6–20.1% | 2.0% | 4.8 | 0.0–4.2% |
| Fruit | 10.5% | 6.9 | 1.3–26.6% | 21.3% | 10.7 | 3.3–42.5% | 22.4% | 17.6 | 14.3–30.6% |
| Insects | 7.2% | 4.7 | 1.1–18.8% | 14.3% | 8.5 | 2.0–34.1% | 6.6% | 9.6 | 2.2–11.1% |
| Game birds | 21.3% | 11.6 | 2.7–44.5% | 13.7% | 8.8 | 1.7–34.5% | 4.4% | 4.9 | 2.1–6.7% |
| Mammals | 52.8% | 15.9 | 18.8–79.6% | 42.0% | 15.6 | 13.6–72.7% | 74.5% | 30.8 | 60.3–88.7% |
| [b]Small mammals | 20.4% | 14.7 | 1.9–55.3% | 17.7 | 12.4 | 2.1–47.5% | 25.8% | 16.4 | 18.2–33.3% |
| [b]White-tailed deer | 32.4% | 17.0 | 3.9–66.3% | 24.3% | 15.8 | 2.8–60.5% | 44.1% | 18.9 | 35.3–52.8% |

Notes:
[a] Frequency of occurrence of primary prey items for 18 coyote packs in SRA during spring/summer of 2016 and 2017; estimated *via* scat analysis. Coyote packs were the unit of replication and values represent food items found in scats of resident coyotes.
[b] Food sources grouped together based on taxonomy (Class level) as "mammalian prey" because they exhibited considerable overlap in isotope space (Fig. 3).

spring/summer especially if they were 9–11-month old pups displaced from natal territories due to hunting and trapping mortalities of breeders. Although approximately 28% (40 out 167) of our radio-collared coyotes changed territorial status during monitoring periods, it is more likely that pups in our study experienced undocumented transitions from residency to transiency in the months prior to their fall/winter capture than did juveniles and adults. Taking this into consideration and our *post-hoc* classification of coyote pups as residents, we believe it is unlikely that we misclassified a substantial number of individuals as either residents or transients. Furthermore, most coyotes (34 of 41) in this study were not long-distance dispersers (>200 km$^2$), indicating coyotes likely continued to use the same areas before and after being radio collared. Therefore, we suggest our findings accurately reflect the observed proportions of agricultural land use and canopy cover and density of roads in space used by SRA coyotes.

## Carbon isotope values

Canopy cover was the only informative predictor in our $\delta^{13}$C GAM in which $^{13}$C enrichment in SRA coyotes declined with increasing canopy cover. This was likely a consequence of predominance of C$_3$ plants in forested, closed-canopy habitats, while more open habitats, such as grasslands, woodlands, and agricultural cover, contain C$_4$ plants, which have greater $\delta^{13}$C (*O'Leary, 1988*; *Ehleringer, Cerling & Helliker, 1997*). Additionally, depleted $\delta^{13}$C values in animals can be attributed to depletion of $\delta^{13}$C values in plants growing under the canopy of dense forest stands (*Drucker et al., 2008*; *Bonafini et al., 2013*; *Sykut et al., 2021*). The depletion of $\delta^{13}$C in plants under closed canopies results from conditions of reduced sunlight, poor ventilation, and greater humidity when compared to those in open habitats (*Teiszen, 1991*; *Sykut et al., 2021*). Approximately 43% (18 of 41) of coyotes included in our GAM inhabited the Thomas-Augusta area and Savannah River Site, the areas with the greatest canopy cover, and exhibited lower $\delta^{13}$C values than did coyotes in the Edgefield-Aiken area. Therefore, when SRA coyotes consume co-occurring

prey species and fruiting plants in areas of increasing canopy cover, they likely experience $\delta^{13}$C depletion resulting from environmental factors influencing isotopic variation at the base of their food chain.

Elevated $\delta^{13}$C values in coyote tissue are generally accepted as an indicator of the prevalence of anthropogenic food sources in coyote diets, especially in urban environments (*Newsome et al., 2010*; *Murray et al., 2015*; *Larson et al., 2020*; *Sugden et al., 2021*; *Gámez et al., 2022*). For example, *Larson et al. (2020)* observed $^{13}$C enrichment in coyotes along a rural-to-urban gradient of southern California and attributed enrichment to increased consumption of anthropogenic foods and domestic cats (*Felis catus*) by coyotes in suburban and urban areas. Several studies used a threshold $\delta^{13}$C value of −20.5‰ to differentiate natural from anthropogenic resources in diets of coyotes (*Newsome et al., 2015*; *Larson et al., 2020*) and other wildlife (*Nicholson & Cove, 2022*). However, a −20.5‰ threshold would indicate that anthropogenic foods were prevalent in 38% of SRA coyote diets (22 of 58 coyotes; Table S1) despite the results of our stable isotopic mixing models herein and a previous scat analysis (*Ward et al., 2018*) strongly contradicting prevalent use of anthropogenic foods by SRA coyotes. In fact, *Kays & Feranec (2011)* reported a mean $\delta^{13}$C value of −20.6‰ (range = −21.6‰ to −18.9‰) for captive canids that were fed a diet of scrap beef and kibbled dog food which is similar to $\delta^{13}$C values in SRA coyotes (mean $\delta^{13}$C = −20.9‰; range = −23.5‰ to −16.2‰; Table S1). Additionally, $\delta^{13}$C values reported in a recent study for 14 of 16 coyotes (Fig. 4C) and all 15 red wolves (*Canis rufus*) sampled from the Albemarle Peninsula of North Carolina (*Arbogast, Hodge & Brenner-Coltrain, 2017*) were above the −20.5‰ threshold despite little evidence of coyotes and red wolves consuming anthropogenic foods (*McVey et al., 2013*; *Hinton et al., 2017*). The exception was the Milltail red wolf pack on Alligator River National Wildlife Refuge that neighbored a county landfill, though their diet comprised mostly wild prey (*Dellinger et al., 2011*). Space used by coyotes on the Albemarle Peninsula consisted of approximately 17% forest cover (*Hinton, van Manen & Chamberlain, 2015*) which was significantly less canopy cover than we observed in SRA. Consequently, elevated $\delta^{13}$C values in coyotes may reflect the broader interplay of canopy cover and other landscape variables, such as agriculture and landscaping, that affect local food web dynamics rather than excessive use of anthropogenic foods by coyotes. Canopy cover can vary considerably across urban land uses such as nature preserves, recreational areas, golf courses, residential and industrial subdivisions, and low- and high-density residential areas (*Porter, Forschner & Blair, 2001*; *Gallo et al., 2017*; *Ossola et al., 2019*) that can affect flora and fauna diversity and facilitate carbon enrichment in food webs along rural-to-urban gradients (*Golubiewski, 2006*; *Larson et al., 2020*; *Rankovic et al., 2020*; *Trammell et al., 2020*).

## Nitrogen isotope values

High $\delta^{15}$N values in coyotes reflects consumption of protein-rich animal prey and increases with carnivory and trophic level (*Reid & Koch, 2017*; *Gámez et al., 2022*). Elevated $\delta^{15}$N values in SRA coyotes was correlated with low road density, intermediate agricultural cover, and reduced space use. Road density was lowest in areas used by coyotes in the Edgefield-Aiken area where agriculture cover was also the greatest. Considering

these two factors and the effect of reduced space use by coyotes, we suggest that carnivory in coyotes likely increases when space use becomes more localized, especially in areas with low human presence and with some agricultural cultivation. Indeed, this appears to support previous findings that consumption of white-tailed deer by SRA coyotes was positively correlated with smaller home ranges (*Ward et al., 2018*) and coyotes were observed avoiding roads and increasing their use of forest-agriculture edges during spring/summer (*Youngmann et al., 2022*). Furthermore, SRA coyotes did not rely on roads for acquiring white-tailed deer *via* scavenging (*Youngmann et al., 2022*) and previous research indicated roads on the Savannah River Site did not increase carrion use by coyotes (*Hill et al., 2018*). However, coyotes were identified as the primary mammalian scavenger of wild pig carcasses on the Savannah River Site (*Turner et al., 2017*) with pigs observed in approximately 8% of coyote scats (see Table 2 in *Schrecengost et al., 2008*). Although most consumption of wild pigs by coyotes on the Savannah River Site occurred after pig control efforts, small hooves were found in scats indicating some predation on wild pigs could not be ruled out (*Schrecengost et al., 2008*). Therefore, elevated $\delta^{15}$N values in SRA coyotes likely resulted from carnivory and some scavenging of wild prey rather than consumption of anthropogenic foods.

Furthermore, coyotes on the Albemarle Peninsula in North Carolina exhibited greater $\delta^{15}$N values than did SRA coyotes (11.2‰ *vs* 7.0‰; Fig. 4C). Agricultural cover comprised approximately 50% of space used by coyotes on the Albemarle Peninsula (*Hinton, van Manen & Chamberlain, 2015*) whereas space used by SRA coyotes consisted of nearly 23% agricultural cover. There are a number of potential explanations for how agricultural land cover can influence $\delta^{15}$N values in coyotes. Firstly, primary prey of coyotes on the Albemarle Peninsula and SRA consisted of white-tailed deer, lagomorphs, and rodents (*Hinton et al., 2017*; *Ward et al., 2018*) and prey fed on agricultural crops which can increase $\delta^{13}$C (*e.g.*, corn) and $\delta^{15}$N (*e.g.*, winter wheat, soybeans, and peanuts). The Albemarle Peninsula was more intensively used for row crops (corn, soybean, and winter wheat) than was SRA in which two of the three primary crops were $C_3$ plants. Secondly, fruit comprised a greater proportion of coyote diets in SRA (~24% of food items in scats; *Ward et al., 2018*) than in diets of those inhabiting the Albemarle Peninsula (<3% of food items in scats; *McVey et al., 2013*; *Hinton et al., 2017*). The substantial differences in agricultural cover and fruit used by coyotes in SRA and Albemarle Peninsula likely attributed to the significant differences in the $\delta^{15}$N values of both populations in which variation in agricultural cover affected the degree of carnivory in coyote populations.

## Coyote carnivory and diet

Coyote diets during spring/summer exhibited significant variability among individuals with mammalian prey and game birds contributing more to resident diets than to transient diets. Although transients appeared to compensate for this by increasing their consumption of fruit and insects, mammalian prey and game birds still contributed to 55.7% of their diet. Current literature detailing the use of white-tailed deer by coyotes throughout the southeastern United States consists of estimates varying between 5% and 68%, *via* scavenging and predation, particularly during spring/summer when neonates

were available (*Schrecengost et al., 2008*; *Kilgo et al., 2012*; *Nelson et al., 2015*; *Hinton et al., 2017*; *Ward et al., 2018*; *Hinton, Rountree & Chamberlain, 2021*; *Youngmann et al., 2023*). Because our samples reflect diets of the previous spring/summer when coyote hairs were last metabolically active (*Castelló, 2018*), our estimated use of white-tailed deer by coyotes likely included their consumption of adults and fawns (*Kilgo et al., 2012*; *Nelson et al., 2015*; *Ward et al., 2018*). For example, *Ward et al. (2018)* reported use of adult white-tailed deer and fawns by resident coyotes in SRA and Alabama during spring/summer months ranged between 13–30% and 11–33%, respectively, with total use of deer ranging between 36–62%. Additionally, recent broad-scale research on coyote diets in the southeastern United States could not identify scavenging behaviors as an explanation for consistent year-round use of white-tailed deer (*Cherry et al., 2016*; *Hinton et al., 2017*; *Ward et al., 2018*; *Youngmann et al., 2022*).

Surprisingly, game birds such as northern bobwhite and wild turkey made significant contributions to the diets of resident and transient coyotes. Although studies typically report game birds as incidental prey for coyotes (*Henke, 2002*; *Staller et al., 2005*; *Schrecengost et al., 2008*; *Melville et al., 2015*; *McVey et al., 2013*), there is some evidence that coyotes may use game birds as a food source in SRA. A recent study using metabarcoding analysis reported that wild turkeys made up 9.2% of prey detected in coyote scats of SRA during spring (*Youngmann et al., 2023*). Therefore, it is possible that *Ward et al. (2018)* underestimated the contributions of game birds in SRA coyote diets; however, they used packs rather than scats as the unit of replication in which consumption of avian prey ranged from 0–14% for packs during spring/summer. Considerable variation in use of game birds by coyote packs (*i.e.*, zero-inflated data) may be responsible for contradictory results produced among studies. Nevertheless, we detected a moderate negative correlation between the contributions of game birds and white-tailed deer in resident ($r = −0.491$) and transient ($r = −0.430$) diets indicating that when individuals consumed more game birds they consumed less deer and *vice versa*. *Youngmann et al. (2023)* observed a similar albeit weak negative correlation between the occurrence of wild turkey and white-tailed deer in coyote scats. Current management practices for game species involving supplemental feeding may cause some game bird populations to concentrate into small areas where they are more susceptible to predation leading to increased variance in game bird use by coyotes. Therefore, we suggest that the observed negative correlations between coyote consumption of white-tailed deer and wild turkeys by both studies indicates prey-switching behaviors in coyotes that permit resident coyotes to maintain stable home ranges (*Ward et al., 2018*).

Small mammals and fruit are commonly reported as substantial food items for coyotes throughout their geographic range (*Jensen et al., 2022*), although work in some regions of the southeastern United States reported that fruit was not an important food item (*McVey et al., 2013*; *Hinton et al., 2017*; *Hinton, Rountree & Chamberlain, 2021*). We observed less consumption of small mammals by coyotes but similar use of fruit than is typically reported by other studies. For example, our study observed less use of small mammals including lagomorphs by coyotes than did the range of mean spring/summer values reported for those prey types in a meta-analysis on coyote diets throughout North America

(*Jensen et al., 2022*). We also observed a lesser proportion of resident coyote diets consisting of fruit than reported by *Ward et al. (2018)* though use of small mammals in our study was on par with those reported in their study. Regardless, our stable isotope mixing models herein and previous scat analysis suggest that the spring/summer diets of SRA coyotes consist of 15–30% small mammals and 10–25% fruit. Fruit seeds are commonly found in coyote scats by which coyotes may serve as an important seed disperser for dominant fleshy-fruited species (*Silverstein, 2005*; *Roehm & Moran, 2013*; *Campbell & West, 2022*; *Draper et al., 2024*). Indeed, fruits comprised a significant proportion of diets for transient coyotes and transients may play an important role in regional dispersal of seeds such as those of blackberries, persimmon, wild plums, and wild grapes more so than residents because of their wide-ranging movements (*Gelmi-Candusso et al., 2023*).

Finally, the contribution of anthropogenic foods that we observed for coyotes in SRA likely resulted from their use of agricultural crops and supplemental feed, as these two anthropogenic food sources are likely more widespread and available to coyotes than is garbage consisting of processed foods. For example, supplemental feeding of white-tailed deer is widely practiced by deer hunters throughout the United States including in SRA to optimize hunting success or enhance antler growth, fecundity, and survival of deer (*Lambert & Demarais, 2001*; *Webb et al., 2008*; *Priesmeyer et al., 2012*; *Sorensen, van Beest & Brook, 2014*; *Haung et al., 2022*). Corn is ubiquitous in supplemental feed for white-tailed deer and game birds and gives feed high $\delta^{13}C$ values (*Nájera-Hillman & Mandujano, 2013*) that may be mistaken for processed anthropogenic food which also contains corn in the form of corn syrup and starch (*Newsome et al., 2010*; *Sugden et al., 2021*). Indeed, macroscopic remains of anthropogenic food such as undigested pasta, food wrappers, and other types of garbage were found in only 0.3% of all scats collected from coyotes in SRA (*Ward et al., 2018*) and foraging behaviors of residents indicated little use of roads and anthropogenic land cover that would expose coyotes to garbage and litter (*Youngmann et al., 2022*). We encourage future studies to consider the effects of agriculture, supplemental feeding of game species, and canopy cover on $\delta^{13}C$ values in coyotes to better understand how coyotes and other canids exploit anthropogenic landscapes and resources. Additionally, in a similar manner that natural prey may be accounted for in diet studies, future study designs should consider surveying study areas and/or coyote territories to identify the distribution and density of predominant sources of anthropogenic foods that study animals are exposed to rather than assuming high $\delta^{13}C$ values are a result of coyote reliance on anthropogenic foods.

## CONCLUSIONS

Stable isotopes are useful in assessing diets of transient coyotes which are typically young, dispersing animals moving extensively across the landscape in search of mates and breeding territories. The diets of these individuals are difficult to quantify using traditional scat analyses but insights can be made by combining radio telemetry and stable isotope analyses. Identifying the territorial status of coyotes sampled for stable isotopes *via* GPS radio-collars, we observed some differences in the proportional contributions of food sources for residents and transients, but no difference in their niche spaces. However, we

observed a canopy effect on $\delta^{13}C$ values and small amounts of anthropogenic food in coyote diets, suggesting that $^{13}C$ enrichment is likely influenced more by land cover features and less by anthropogenic food sources. Additionally, agricultural cover, road density, and space use behaviors of coyotes influenced $\delta^{15}N$ values, indicating that land cover and coyote residency can influence degree of carnivory. Given the reliance of our interpretations of coyote feeding habits derived from previous work, we offer that our study will serve to focus future research to develop more comprehensive study designs for studying coyote diets rather than favoring one particular method over another.

## ACKNOWLEDGEMENTS

We appreciate assistance in trapping by D. Eaton and R. Johnson. We are grateful to many private landowners that granted us access to their properties for this research. This manuscript was prepared as an account of work sponsored by an agency of the United States Government. Neither the United States Government nor any agency thereof, nor any of their employees, makes any warranty, express or implied, or assumes any legal liability or responsibility for the accuracy, completeness, or usefulness of any information disclosed, or represents that's its use would not infringe privately owned rights. Reference herein to any specific commercial product, process, or service by trade name, trademark, manufacturer, or otherwise does not constitute or imply its endorsement, recommendation, or favoring by the United States Government or any agency thereof. The views and opinions of the authors expressed herein do not necessarily state or reflect those of the United States Government or any agency thereof. The findings and conclusions in this article are those of the authors and do not necessarily represent the views of the Alabama Department of Conservation and Natural Resources, Georgia Department of Natural Resources—Wildlife Resources Division, the South Carolina Department of Natural Resources, or the US Department of Energy. We thank J. Gittleman, J. Kilgo, and N. Nibbelink for their review of an earlier version of this manuscript.

### Funding

Funding for this study was provided by a Federal Wildlife Restoration grant, Alabama Department of Conservation and Natural Resources, Georgia Department of Natural Resources—Wildlife Resources Division, South Carolina Department of Natural Resources, and Warnell School of Forestry and Natural Resources at the University of Georgia. Funding was also provided by the U.S. Department of Energy under Award No. DE-EM0004391 and DE-EM0005228 to the University of Georgia Research Foundation. Contributions of James C Beasley and Sarah C Webster were partially supported by the U.S. Department of Energy Office of Environmental Management (award number DE-EM0005228 to the University of Georgia Research Foundation). The funders had no role in study design, data collection and analysis, decision to publish, or preparation of the manuscript.

## Grant Disclosures

The following grant information was disclosed by the authors:

Federal Wildlife Restoration.

Alabama Department of Conservation and Natural Resources.

Georgia Department of Natural Resources—Wildlife Resources Division.

South Carolina Department of Natural Resources.

Warnell School of Forestry and Natural Resources at the University of Georgia.

U.S. Department of Energy: DE-EM0004391, DE-EM0005228.

University of Georgia Research Foundation.

## Competing Interests

The authors declare that they have no competing interests.

## Author Contributions

- Sarah C. Webster conceived and designed the experiments, performed the experiments, analyzed the data, authored or reviewed drafts of the article, and approved the final draft.
- Joseph W. Hinton conceived and designed the experiments, performed the experiments, analyzed the data, prepared figures and/or tables, authored or reviewed drafts of the article, and approved the final draft.
- Michael J. Chamberlain conceived and designed the experiments, authored or reviewed drafts of the article, and approved the final draft.
- Jazmin J. Murphy analyzed the data, authored or reviewed drafts of the article, and approved the final draft.
- James C. Beasley conceived and designed the experiments, authored or reviewed drafts of the article, and approved the final draft.

## Animal Ethics

The following information was supplied relating to ethical approvals (*i.e.*, approving body and any reference numbers):

University of Georgia Institutional Animal Care and Use Committee

## Field Study Permissions

The following information was supplied relating to field study approvals (*i.e.*, approving body and any reference numbers):

Animal captures and handling were approved by the Georgia Department of Natural Resources, South Carolina Department of Natural Resources, and Alabama Department of Conservation and Natural Resources.

## Data Availability

The data are available in the Supplemental Files.

## Supplemental Information

Supplemental information for this article can be found online at http://dx.doi.org/10.7717/peerj.17457#supplemental-information.

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
