# Peer review of "Land cover and space use influence coyote carnivory: evidence from stable-isotope analysis"

_PeerJ, doi:10.7717/peerj.17457_

## Round 0.1 · original submission · Major Revisions

All reviewers and myself appreciated your work. The reviewers made very constructive comments (that encompass my comments) that I would like you to consider in a revised document.

Reviewer 1 ·

Basic reporting

This study describes dietary differences between resident and transient coyotes in the Southeastern United States based on diet data from stable isotopes and home range data from GPS collars. Overall this study is clear and well-written and focuses on an interesting question with a useful combination of data types. All basic reporting is at a professional level.

Experimental design

L98 – 101: Did you have a hypothesis about comparing diet breadth between residents and transients? For example, might transients have greater diet breadth (larger dietary niches) because they have less access to preferred foods? Also, I am surprised that you predicted that residents would have more deer in their diet but lower 15N signatures. I think that needs more set up in the introduction.

L240 – 242: I believe washing the hair in 2:1 chloroform:methanol (or another solution) is intended to remove oil from the hair and is standard practice. Could you provide support for not removing the oils?
“lipids are depleted in 13C compared with protein, and it is consequently a routine to remove lipids prior to measuring carbon isotopes in ecological studies so that variation in lipid content does not obscure variation in diet.” Elliott, K.H., Roth, J.D. and Crook, K., 2017. Lipid extraction techniques for stable isotope analysis and ecological assays. Lipidomics: methods and protocols, pp.9-24.

L272: Looking at Figure 2, are you likely to differentiate between deer and rodent consumption if they are so isotopically similar?

Validity of the findings

The data have been provided, statistical analysis is rigorous, conclusions are well-supported by the results.

Additional comments

I would like to see more explanation of the broader implications of this study – what is the utility of these results for either informing coyote ecology or management? This would be appropriate in the conclusions section and at the end of the abstract.

Discussion:
L324-328: Could an age class comparison help resolve this?


Minor comments
L117 – 118: I find this description kind of confusing. I would suggest adding “pine” before “plantations” and remove the italics on plantations.
L125 – 127: I suggest reminding readers that this is the study using scats within home ranges and limited to residents.
L167: I would suggest specifying you mean the ratio of 15N/14N and 13C/12C
L293 – 295: I would suggest moving the results about the size of the isotopic niche space (L300-303) here.
L272: I would suggest reminding the readers that the food sources are deer, rodents, fruit, and anthropogenic food.

·

Basic reporting

This manuscript easily exceeds all the criteria associated with basic reporting: 1) it is written clearly and conforms to professional standards of courtesy; 2) the knowledge gap regarding diet breadth of transient versus resident animals is well-articulated and grounded in previous literature; 3) the article structure is easy to follow, the included figures support the text and are well-described and labeled, and the raw data have been made available; and finally, 4) the submission is an appropriate unit of publication that includes results that pertain to the two stated hypotheses (i.e., that resident and transient coyote diet breadth would largely overlap, but that transient coyotes would consume more anthropogenic resources, and that resident coyotes would consume more white-tailed deer).

I thank the authors for providing their raw data, however it would nonetheless be helpful if additional data and metadata were included in the supplemental data file to make it more useful to future readers. At present, it is unclear what the group and community variables delineate. Further helpful additions would be: 1) the %C and %N values (used to account for concentration dependence) for the food resources in the mixing models, 2) more information about the animals and fruit (e.g., species, location collected) and the rest of the coyote data (e.g., sex, age, location trapped, date sampled, etc.). Note also that the labels iso1 and iso2 would be more communicative if replaced with d13C and d15N. Finally, it would be helpful to know if the isotope values are as measured or adjusted for trophic discrimination and/or any tissue discrimination (e.g., difference between hair and muscle for deer and rodents).

Experimental design

First of all, this submission represents original primary research within the Aims and Scope of the journal. Secondly, the research question is well defined (are there differences in diets between resident and transient coyotes?). By combining spatial and dietary data from individual coyotes, this study will be able to address the above question and fill an existing knowledge gap regarding how space use affects dietary composition. Thirdly, the research was performed in conformity with ethical standards for the field (all permit information is documented in the manuscript) and sample collection was rigorous.

I commend the authors for thoughtfully designing their study. I do, however, think the communication of their methods could be improved. Firstly, given that the space use analysis was previously published in Webster et al. (2022), I recommend re-writing this paragraph (Lines 222-237) such that it is more clear that this work was done for a previous publication. If there are differences between Webster et al. (2022) and the analysis being described here, then those differences should be highlighted and explained. Secondly, the description of the isotopic analyses could be more complete – which standards were used for QAQC (please also define this term) and what is the error on the measurements? How were the samples corrected to the air and PDB scales and which standards were used for this purpose? Bond and Hobson (2012) provide clear guidelines for reporting stable isotope analysis results. Lastly, the diet-to-hair fractionation factor quoted is incorrect – Roth and Hobson (2000) report a difference of 2.6 per mil for d13C. It may be that the authors are trying to account for the fact that they’ve measured isotope values in animal hair rather than muscle (e.g., Newsome et al. 2015), but this should be explicitly stated. Further, other dietary items like fruit and the estimated anthropogenic food resources don’t require for this tissue difference to be accounted for. My suggestion is to correct the hair to muscle for rodents and deer and then use the published fractionation factor for the analysis.

Bond, A.L., Hobson, K.A., 2012. Reporting stable-isotope ratios in ecology: recommended terminology, guidelines and best practices. Waterbirds 35(2):324-32.

Newsome, S.D., Garbe, H.M., Wilson, E.C. and Gehrt, S.D., 2015. Individual variation in anthropogenic resource use in an urban carnivore. Oecologia, 178(1):115-128.

Validity of the findings

Again, I commend the authors for thoughtfully designing a study that allows for both diet and space use to be evaluated – this is no easy task. My main concern, however, is that the authors’ conclusions about coyote diet are based on a modeling approach that does not follow best practices (see Phillips et al., 2014 for an excellent guide). First, when using stable isotope data to estimate the proportional contributions of different dietary sources to diet as a whole, the dietary sources need to have distinct isotopic signatures (Gannes et al. 1998). This is not the case for the rodent and deer data presented in this study – as can be seen in Figure 2, these two diet sources are completely overlapping in isotope space. No model will be able to distinguish between these two sources; presently, the model is swapping one for the other in the solutions, such that both have diffuse probability distributions, as is evident in Figure 4. Mixing models are also “sensitive to missing sources and may produce erroneous results if all sources are not included; hence it is critical that all food sources are sampled” (Phillips et al. 2014). It is clear that diet sources are missing from this analysis because 22 (out of 43) of the coyotes plotted in Figure 2 fall outside of the mixing space created by the presently considered dietary sources. Furthermore, previous research in the area showed that coyotes had a huge variety of small mammals in their diets (Ward et al. 2018), of which only two are presently being considered. The best way to address these issues will be to collect more isotope data from potential coyote dietary sources. Rather than capturing animals to accomplish this, the authors could potentially source hair samples from natural history collections or even from coyote scats (e.g., bone samples from small mammals can be readily identified and then processed for isotopic analysis). There may also be isotope data available in the literature. Alternatively, the mixing model analysis could be dropped from the study and the authors could instead focus on coyote dietary breadth.

Gannes L.Z., Martínez del Rio C., and Koch P. 1998. Natural abundance variations in stable isotopes and their potential uses in animal physiological ecology. Comp. Biochem. Physiol. A Mol. Integr. Physiol. 119(3): 725–737.

Phillips, D.L., Inger, R., Bearhop, S., Jackson, A.L., Moore, J.W., Parnell, A.C., Semmens, B.X. and Ward, E.J., 2014. Best practices for use of stable isotope mixing models in food-web studies. Canadian Journal of Zoology 92:823-835.

Additional comments

Please find below a few minor comments/questions associated with specific lines of text.

Lines 65-68: Could you speak to whether or not there has been any previous research into the demographics of transient versus resident coyotes? Are transients more likely to be male? Or more likely to be a particular age?

Lines 98-101: Could you be more explicit about why you think white-tailed deer will be more important in the diets of resident coyotes?

Lines 125-133: Could you please communicate here what technique was used by Ward et al. (2018) to estimate coyote diets? Also, this is a much more diverse set of food resources than was sampled for this study – as mentioned above, I think the study would be improved with additional diet samples from specifically from other small mammals.

Lines 158-164: Thank you for so clearly communicating this information about all of your permits.

Lines 204-207: This sentence is a bit confusing as written, I suggest revising to: “We estimated the isotopic composition of anthropogenic food resources by averaging isotopic ratios of North American human hair from several studies (list them) and accounting for trophic fractionation (references)." Or something along those lines.

Lines 292-293: Are there any age differences between resident and transient coyotes?

Lines 303-304: The supplementary data table includes 4 rodents, not 3.

Line 340: replace “its” with “their”, given that “coyotes” are being discussed.

Table 1: The inclusion of the %C and %N data would be helpful here. Also, given the usual error on isotope ratio measurements, it would be better to report these results out to just 1 or 2 decimal points, not 3.

Table 2: Would it make sense to report the data from Ward et al. (2018) in the same format as those in this study – so 0.241 instead of 24.1?

Figure 1: This map is fantastic.

Figure 3: I think it’s fascinating that essentially two transient coyotes are responsible for the difference between the groups – could you discuss those individuals in the text? Is there anything special about them?

Reviewer 3 ·

Basic reporting

No comment

Experimental design

a. 1) I think the predictions presented by the authors are valid. However, they were not supported by the introduction through logical progression statements supported by the literature. For example, the predictions explicitly mention differences among resident and transient coyote diets relative to anthropogenic foods and white-tailed deer, yet neither of these food items were mentioned in prior to the predictions. I think this could be improved by adding an additional paragraph that supports the logic behind these predictions.

Validity of the findings

2) There was considerable individual variation in diet within resident and transient coyotes. The authors acknowledge this in the manuscript. But occasionally this variation is not adequately reported. For example, in the abstract, the authors wrote:

"We found that transient coyotes consumed greater proportions of rodents compared to residents (29.7% vs 20.3%), and transient diet composition varied considerably among individuals. Resident diets primarily consisted of white-tailed deer (39.7% ± 15.1%), anthropogenic foods (24.8% ± 5.1%), and rodents."

In the first sentence the authors do not describe the variation around these mean values. In the second sentence the authors report some measure of variation but do not specify whether this is SD or SE or some other measure of variance. The same +- notation is used throughout the results and it is also unclear what measure of variance is being referenced.

3) Why was US state considered a random effect? Does this mean something ecologically or statistically?

4) I am not an expert in stable isotope diet analysis, but it appears that the isotopic signatures of deer and rodents have a high degree of overlap. How might this overlap have impacted the interpretation of the findings relative to rodent and deer diet among residents and transients? It is difficult for me to imagine how these two isotopic signatures could be reliably distinguished and if they cannot then the interpretation of differences in diet between groups relative to rodents and deer should probably be tempered.

5) Several recent articles that have examined coyote space use recognize three distinct space use strategies instead of the binary (resident and transient). See Morin et al 2017, Sasmal et al 2019, Ellington et al 2020. I think it is reasonable to assume that these three space use strategies might have different diets due to differences in familiarity with the landscape and energetic needs. Transient coyotes that are waiting for high quality territories (biding) or local transients would be expected to be more familiar with their landscape than transients that are moving long-distances and rarely reusing sites. I think the authors should consider evaluating coyote diets across the three different strategies or at a minimum discuss how a 'hidden' space use strategy within their data might have impacted the interpretation of their results.

6) The disconnect between the stable isotope window and the monitoring of space use (and thus the distinction of space use strategy) is an important limitation. I think it would be informative if the authors reported summary statistics on the time between GPS monitoring and the isotopic window (average, SD, range, etc.) or better yet a complete table with data for each individual. Furthermore, the implications of this limitation could be discussed more in depth in the discussion.

Morin, D. J., & Kelly, M. J. (2017). The dynamic nature of territoriality, transience and biding in an exploited coyote population. Wildlife Biology, 2017(1), 1-13.

Sasmal, I., Moorman, C. E., Swingen, M. B., Datta, S., & DePerno, C. S. (2019). Seasonal space use of transient and resident coyotes (Canis latrans) in North Carolina, USA. Canadian Journal of Zoology, 97(4), 326-331.

Ellington, E. H., Muntz, E. M., & Gehrt, S. D. (2020). Seasonal and daily shifts in behavior and resource selection: how a carnivore navigates costly landscapes. Oecologia, 194(1-2), 87-100.

Additional comments

7) The authors discussion of the fruit as a dietary item for coyotes could be enhanced by discussing their findings relative to a recent paper about coyotes as a vector for seed dispersal in urban landscape. (see Gelmi-Candusso et al. 2023).

8) The color choices in figure 2 make it difficult to distinguish fruit, rodents, and deer. It is unclear whether the residents and transients are meant to be different colors. Finally, this image appears pixelated, and the authors should ensure that high resolution figures are used for the publication.

9) I wonder if it would be informative to plot the estimated proportion of different diet items for each coyote to better visualize the large degree of variation within space use strategy. It could be a 4-panel figure, one panel for each diet item.

Gelmi-Candusso, T. A., Wheeldon, T. J., Patterson, B. R., & Fortin, M. J. (2023). The effect of urbanization and behavioral factors on coyote net displacement and its implications for seed dispersal. Urban Ecosystems, 1-11.

---

## Round 0.2 · Minor Revisions

Thank you for your revised version. All reviewers appreciated the consideration you made of their comments. Reviewer 2 still has some comments that I would like you to address before making my final recommendation.

I also have minor comments:

L204-205. It gives the impression that age can only be assessed in categories while in fact you can age the animals to the year or even months? Can you clarify how those 3 age categories allow you to calculate age to the Months on L358-359?

L265. I would suggest describing how you collected the food item before mentioning how they were analyzed because the reference to fruit here is lacking context.

Finally, I am also wondering if you should consider putting one of your gam model figure into the main text to support your results. In any case, I would suggest improving the resolution of those figures for final publication.

Reviewer 1 ·

Basic reporting

Thank you for revising the manuscript. I have no further concerns and the discussion of the dietary gradient from omnivory to carnivory within coyotes is interesting.

Experimental design

The research question is well defined and fills a knowledge gap. Methods are described well.

Validity of the findings

Findings are valid.

·

Basic reporting

This manuscript still easily exceeds all the criteria associated with basic reporting. I thank the authors for adding additional data to their supplementary data tables.

Experimental design

In general, the authors have done an excellent job of addressing previous comments. I thank the authors for including error estimates for their isotopic analyses; it would still be helpful to know how these error estimates were calculated. Additionally, it is good practice to include information about how samples were corrected to the AIR and VPDB scales and which standards were used for this purpose (e.g., two-point linear normalization using USGS 40 (ẟ13C =-26.39, ẟ15N = -4.52) and USGS 41 (ẟ13C = 36.55, ẟ15N = 47.55)) – doing this will ensure data useability in the future (Szpak et al. 2017). I appreciate that the authors have corrected their trophic fractionation factors and simplified the estimation of anthropogenic resources, however I do still think it important to adjust the deer and small mammal hair isotope data to muscle, which is the tissue the coyotes would actually be assimilating into their own tissues. I suggest the approach taken by Newsome et al. 2015.

Newsome, S.D., Garbe, H.M., Wilson, E.C. and Gehrt, S.D., 2015. Individual variation in anthropogenic resource use in an urban carnivore. Oecologia, 178(1):115-128.

Szpak, P., Metcalfe, J.Z. and Macdonald, R.A., 2017. Best practices for calibrating and reporting stable isotope measurements in archaeology. Journal of Archaeological Science: Reports 13:609-616.

Finally, I was surprised by the new addition of C/N ratios in the GAM analysis; I think this is misguided. The authors mention on lines 129-131 that “Also, carbon-to-nitrogen (C/N) ratios can serve as an additional indicator of coyote carnivory as a largely plant-based diet may produce high C/N ratios (Reid & Koch, 2017)” but this is not true of coyote hair samples – this observation was specific to coyote scats, which contain a lot of undigested materials. I would expect coyote hair to have C/N values that are all quite similar to each other reflecting values expected for hair keratin (e.g., between ~3-4). I think this part of the analysis should be removed.

Validity of the findings

It’s fantastic that the authors have added more potential coyote dietary resources to their analysis – I think the solution of citing values from the literature is a good one in this case. It would be helpful to briefly discuss these literature values and their sources in the main text rather than completely relegating that information to the supplementary material – given that isotope values can vary quite significantly geographically, it would be helpful to be told where the data are sourced from (e.g., game bird data from the southeast will be more relevant than data from elsewhere in the US). And I suggest excluding any data from pen-raised or farm-raised animals, as their diets will not necessarily reflect a natural diet. I also noticed that some of the new data look like they come from individuals consuming C3 resources while other individuals look like they are consuming C4 resources – averaging values across these two end members may not be all that informative. It might be worthwhile to consider breaking apart the groups by closed (C3) versus open (C4) habitat. However you end up summarizing them, the addition of these new sources resolves the issue of missing dietary sources. It does not, however, resolve the issue of deer and small mammals overlapping in isotope space. I see that the authors discuss a combined “mammalian prey” category, but the results presented in the tables include deer and small mammals as separate categories. It’s tempting to try and separate these prey resources, as they represent significantly different size classes of prey, but they are not distinguishable isotopically and the results and discussion should focus on the combined “mammalian prey” category.

Additionally, I appreciate that the authors have added information about the ages of the coyotes to their results and discussion. This information introduces a new issue, however, and that is the fact that nursing animals have elevated carbon and nitrogen isotope ratios because they are essentially consuming their mother’s tissues (Fuller et al. 2006). If there are pups included in the analysis that are young enough to have been nursing during the spring/summer when the sampled hair was growing, then these individuals ought to be excluded from the analysis. In looking at the supplementary data, it’s clear that the two individuals with very high ẟ13C and ẟ15N values are just 9-months old and candidates for exclusion for this reason. I touch on this in a bit more detail below in the line-by-line comments.

Fuller, B.T., Fuller, J.L., Harris, D.A., and Hedges, R.E.M., 2006. Detection of breastfeeding and weaning in modern human infants with carbon and nitrogen stable isotope ratios. American Journal of Physical Anthropology 129(2): 279-293.

Additional comments

Please find below a few minor comments/questions associated with specific lines.

Line 36: Replace “ẟ13C enrichment” with “13C enrichment” because the word “enrichment” is in reference to a particular isotope, in this case the heavy isotope 13C (Sharp, 2017; see table 2.1).

Sharp, Z. 2017. Principles of stable isotope geochemistry.

Lines 129-131: Again, this is specifically in reference to coyote scat, not hair. I would delete this and any analysis of C/N ratio going forward.

Lines 273-278: Again, more information about the standards would be helpful here (e.g., Bond et al. 2012, Szpak et al. 2017).

Bond, A.L., Hobson, K.A., 2012. Reporting stable-isotope ratios in ecology: recommended terminology, guidelines and best practices. Waterbirds 35(2):324-32.

Szpak, P., Metcalfe, J.Z. and Macdonald, R.A., 2017. Best practices for calibrating and reporting stable isotope measurements in archaeology. Journal of Archaeological Science: Reports 13:609-616.

Lines 351-353: I suggest also correcting small mammal and deer hair to muscle values, as muscle is the tissue that coyotes are consuming and assimilating (e.g., Newsome et al. 2015).

Newsome, S.D., Garbe, H.M., Wilson, E.C. and Gehrt, S.D., 2015. Individual variation in anthropogenic resource use in an urban carnivore. Oecologia, 178(1):115-128.

Lines 361-365: This is helpful information. Pups that are primarily consuming milk will look very different isotopically. Looking at your data in Figure 3, it appears that two of the pups in particular have really high ẟ13C and ẟ15N values. I suggest that these two individuals be excluded from your dietary analysis – they fall outside of the mixing space created by potential dietary resources because they are still deriving the majority of their diet from nursing.

Line 384: Replace “spaced” with “space”.

Line 388: Replace “spaced” with “space”.

Line 394: Replace “ẟ13C depleted” with “13C depleted”.

Line 397: Replace “ẟ15N enriched” with “15N enriched”.

Lines 418-419: Given that deer and small mammals are not separable isotopically, it would be better not to include them separately here and instead, Table 3 should simply include results for the combined “mammalian prey” category.

Lines 425-431: The same goes for Table 4 – it should include results for the combined “mammalian prey” category, not deer and small mammals separately.

Lines 433-442: I’m confused by this paragraph – how are these results different from those presented in the previous paragraph? These are probabilities of an individual having particular resources in their diet rather than the predicted proportional contribution of particular resources to diet? Is this just another way of communicating the mixing model results?

Line 492: Replace “ẟ13C enrichment” with “13C enrichment”.

Lines 493-496: Is it necessary to invoke the canopy effect here? Couldn’t it just be a difference of C3 versus C4 plant presence on the landscape? Forested, closed-canopy habitats primarily contain C3 plants, while more open habitats, such as grasslands and woodlands, contain some C4 grasses, which have higher ẟ13C values (e.g., O’Leary 1988; Ehleringer et al. 1997).

O’Leary, M.H. (1988) Carbon isotopes in photosynthesis. BioScience 38(5), 328-336.

Ehleringer J. R., Cerling T. E., and Helliker B. R. (1997) C4 photosynthesis, atmospheric CO2, and climate. Oecologia 112, 285–299.

Line 506: Replace “Enrichment of ẟ13C is” with “Elevated ẟ13C values in coyote tissues are”.

Line 537: Replace “Enrichment of ẟ15N” with “Elevated ẟ15N values”.

Lines 541-542: It’s also possible that the two pups with elevated values are driving this observation – I would remove them from the analysis and re-evaluate whether or not this observation still holds true.

Line 554: Replace “ẟ15N enrichment” with “15N enrichment”.

Line 575: Again, this is true in coyote scat, which contains undigested food, not in hair. I would remove this.

Line 636: Replace “its” with “their”.

Line 642-645: Is it possible that Ward et al. (2018) were looking at coyote diets during a different season? I would imagine that fruit is a seasonally important dietary item and not necessarily consumed in such large proportions throughout the year, as it is only seasonally available in significant quantities.

Line 682: Replace “ẟ13C enrichment” with “13C enrichment”.

Reviewer 3 ·

Basic reporting

no comment

Experimental design

I appreciate the edits made to the manuscript.

Validity of the findings

I think the changes made by the authors have improved the manuscript.

---

## Round 0.3 · accepted · Accept

Thank you for addressing the comments. I assessed the revision myself, and I am happy with the current version.